# Reliability of gamified reinforcement learning in densely sampled longitudinal assessments

Monja P. Neuser[1]☯, Anne Kühnel[1,2,3]☯, Franziska Kräutlein[1], Vanessa Teckentrup[1,4], Jennifer Svaldi[5], Nils B. Kroemer [1,4,6]*

1 Department of Psychiatry and Psychotherapy, Tübingen Center for Mental Health, University of Tübingen, Tübingen, Germany, 2 Department of Translational Psychiatry, Max Planck Institute of Psychiatry and International Max Planck Research School for Translational Psychiatry (IMPRS-TP), Munich, Germany, 3 Section of Medical Psychology, Department of Psychiatry & Psychotherapy, Faculty of Medicine, University of Bonn, Bonn, Germany, 4 School of Psychology & Trinity College Institute of Neuroscience, Trinity College Dublin, Dublin, Ireland, 5 Department of Psychology, Clinical Psychology and Psychotherapy, University of Tübingen, Tübingen, Germany, 6 German Center for Mental Health, Tübingen, Germany

☯ These authors contributed equally to this work.
* nkroemer@uni-bonn.de

**Data Availability Statement:** Run-based data, including maximum-likelihood estimates of the reported results are publically available on osf: https://osf.io/92s4d/

## Abstract

Reinforcement learning is a core facet of motivation and alterations have been associated with various mental disorders. To build better models of individual learning, repeated measurement of value-based decision-making is crucial. However, the focus on lab-based assessment of reward learning has limited the number of measurements and the test-retest reliability of many decision-related parameters is therefore unknown. In this paper, we present an open-source cross-platform application Influenca that provides a novel reward learning task complemented by ecological momentary assessment (EMA) of current mental and physiological states for repeated assessment over weeks. In this task, players have to identify the most effective medication by integrating reward values with changing probabilities to win (according to random Gaussian walks). Participants can complete up to 31 runs with 150 trials each. To encourage replay, in-game screens provide feedback on the progress. Using an initial validation sample of 384 players (9729 runs), we found that reinforcement learning parameters such as the learning rate and reward sensitivity show poor to fair intra-class correlations (ICC: 0.22–0.53), indicating substantial within- and between-subject variance. Notably, items assessing the psychological state showed comparable ICCs as reinforcement learning parameters. To conclude, our innovative and openly customizable app framework provides a gamified task that optimizes repeated assessments of reward learning to better quantify intra- and inter-individual differences in value-based decision-making over time.

## Author summary

Learning from rewards is a fundamental aspect of motivation and alterations in learning and value-based choices are evident across different mental disorders. However, the traditional lab-based assessments provide only limited measurements, hindering our

**Funding:** The study was supported by the Else Kröner-Fresenius Stiftung, grant 2017_A67, the Deutsche Forschungsgemeinschaft (DFG) grant KR 4555/9-1 and KR 4555/10-1 (all granted to NBK), and the Wikimedia foundation, open science scholarship (granted to VT). VT & NBK received salary support from the University of Tübingen, Faculty of Medicine fortune grant #2453-0-0. MPN received additional salary support from the University of Tübingen, Faculty of Medicine 'forschungsorientierte Gleichstellungsförderung' 2605-0-0 awarded to NBK. The funders had no role in study design, data collection and analysis, decision to publish, or preparation of the manuscript.

**Competing interests:** The authors have declared that no competing interests exist.

understanding of potential changes in learning and decision making over time and their association with mental health. To overcome this limitation, we developed an open-source application called Influenca. It combines a new reward learning task with assessments of mental and physiological states, allowing for repeated measurements over weeks. The task involves identifying the most effective medication by considering rewards and changing probabilities of winning and participants receive in-game feedback on their progress. In this validation study, we dissect variability in reinforcement learning parameters within and between individuals, highlighting the importance of repeated assessments for clinical applications. Crucially, we show that the quality of the measurement improves over runs as indicated by a higher test-retest reliability of differences in behavior. In conclusion, the Influenca app offers a gamified task that empowers researchers to better track individual changes in value-based decision-making over time. By utilizing this tool, users can gain insights into aberrant decision-making processes in mental disorders and potentially monitor the effects of interventions.

## Introduction

Learning from past experiences is essential to optimize decision-making and adaptive behavior. Reinforcement learning models provide useful quantifications of individual choice behavior and the integration of information over repeated decisions [1]. Disturbances in reward learning may result in maladaptive choices which have been linked to various mental and metabolic disorders, such as depression [2–4], eating disorders [5], and obesity [6,7]. Parameters of individual reinforcement learning, such as the learning rate or reward sensitivity may even serve as transdiagnostic biomarkers [8] for aberrant cognitive processes that contribute to key symptoms of disorders, such as apathy or anhedonia [9,10]. In light of the growing interest in reinforcement learning for psychological diagnostics, it is worth noting that the effective use of measures as biomarkers for prediction and classification of mental function requires a thorough evaluation of their psychometric properties [11–13]. However, since most studies are conducted in laboratory settings with a limited number of participants and repeated assessments, a systematic evaluation of the psychometric properties of reinforcement learning parameters, such as their test-retest reliability, is still lacking.

To overcome practical limitations of scale in lab-based testing, online and smartphone-based assessments are becoming increasingly popular. They enable the acquisition of large datasets across multiple time points while participants go about their daily lives [14], thereby improving generalizability to robust behavioral predictions outside of the laboratory. In mental health research, methods such as ecological momentary assessment (EMA) or experience sampling are increasingly common to monitor fluctuations in mood or other psychological and physiological states [e.g., 15,16,17,18,19,20]. Consequently, tracking fluctuations in mental states over time may help predict the onset of disorder-specific behavior which is impossible to recreate in the lab, such as binge eating [19,21,22] or binge drinking [23–25]. Beyond practical aspects of data collection, smartphone-based assessments may also reach a more diverse population of users, which increases the variance between participants and improves generalizability [14,26]. Reinforcement learning tasks have been implemented in online formats before, showing the merit of big samples which led to revisions in commonly used models [27,28]. However, previous implementations do not include repeated assessments. To conclude, online assessments of reinforcement learning may provide a powerful means to collect ecologically and psychometrically valid estimates to predict individual trajectories if they allow for the repeated collection of decision-related parameters over time.

To interpret individual trajectories, including markers of clinical progression or potential effects of interventions [29], a sufficient test-retest reliability is necessary. Consequently, the lack of formal assessments of reliability might hamper the widespread application of reinforcement learning tasks to study associations with psychopathology [30,31]. Reliability levels for decision-making tasks typically vary from low to moderate [32]. In reinforcement learning tasks estimates of computational models also exhibit low to moderate reliability [33–35]. Tasks related to probabilistic reinforcement learning have shown comparably low reliabilities in clinical samples as well [36,37]. Preliminary evidence has shown that reliabilities do not differ between online and lab-based assessments [38] or between raw dependent variables and latent variables [39]. Furthermore, test-retest reliability does not only depend on the task itself but may be improved by careful experimental planning [31,40] or hierarchical modeling approaches, such as hierarchical Bayesian estimations [41], integrating repeated measures into the generative model [42,43]. Likewise, integrating response times in addition to choices may improve reliability [44], although such approaches are not commonly used in clinical samples yet. Taken together, despite the widespread use of reward learning tasks, the reliability of individual differences related to value-based decision-making and reward learning is still underreported, especially across many runs in naturalistic settings [45] that would enable monitoring of individual disease trajectories at scale.

To summarize, cross-sectional measures provide a snapshot of value-based decision-making, which cannot separate trait-like differences from state-like differences in behavior, specifically if reliability is low. To provide a much more comprehensive estimation of reliability in a clinically relevant setting, we present densely sampled longitudinal data of reinforcement learning and decision-making in a sample with diverse symptoms of psychopathology (open-source app Influenca: www.neuromadlab.com/en/influenca-2). To facilitate the investigation of altered value-based decision making in participants with pathological eating behavior (e.g., restrictive eating or binge eating) or mood and anxiety disorders, we developed a smartphone-based, gamified assessment. We measured decision-making over extended time periods in a large sample recruited to span a large range of body mass index (BMI) and with diverse symptoms of psychopathology. The longitudinal assessment across 30 runs enables us to investigate how reinforcement learning changes over runs and how variable behavior is within participants. Such a dense sampling approach provides important new information beyond a lab-based snapshot of behavior to characterize individual trajectories of behavior. For example, binge eating might be characterized by increased fluctuations in reward sensitivity are associated with higher fluctuations in reward learning [46]. Here, we first evaluate the psychometric properties, such as reliability of behavioral parameters to ensure that the data provided by the app is able to provide reliable trait-like inter-individual differences while still capturing meaningful intra-individual variability. Such intra-individual variability may reflect potential adaptations in individual behavior, for example, in response to treatments, reflecting the range of the individual repertoire of behavior. Hence, more detailed knowledge of the psychometric characteristics of these parameters will provide the foundation for future in-depth characterization of aberrant value-based decision-making in mental disorders in the future.

## Methods

### Participants

The initial sample included 648 individuals who downloaded and played Influenca between April 2019 and 1$^{st}$ of July 2021. Of these, 391 participants (60%) completed at least 10 runs and 235 participants (37%) completed all 31 runs ($M_{runs}$ = 16.7, SD = 12.5, S1 Fig). Of note, 42 participants (10%) started the game before March 2020, whereas the majority only started playing

after the onset of COVID-19 pandemic. Still, participants starting before onset of the COVID-19 pandemic showed similar changes in behavior across runs compared to participants starting later (see S1 Appendix). To estimate changes in reinforcement learning parameters and their test-retest reliability across repeated runs, we included participants who completed at least 10 runs of Influenca after passing quality control (i.e., 106 runs were excluded due to random choices: log-likelihood < -100.13, S2 Fig) in the current analysis. This led to a final sample of N = 384 participants ($M_{age}$ = 35.78 years, $SD$ ± 14.13, 291 women, S1 Table) with 9729 valid runs. The sample was recruited across multiple studies and included participants with a wide range of BMI (14–58, MBMI = 26.4 $kg^2$/m, SD ± 7.4) that was enriched for pathological eating and mood and anxiety disorder. Consequently, 43 participants fulfilled the criteria for a binge eating disorder and 107 participants reported symptoms of depression according to suggested cutoffs for the Beck Depression Inventory II (> = 14, S1 Table, BDI) [47].

This study was performed in line with the principles of the Declaration of Helsinki. Approval was granted by the Ethics Committee of University Tübingen (09.08.2017 / No. 393/2017BO2). Informed consent was obtained twice. First, all participants provided informed consent by clicking a checkbox [48] when registering for Influenca, stating they agree with the terms of service and usage of pseudonymized and anonymized data for the specified scientific objectives. Second, participants were provided with a second informed consent form before completing online questionnaires after completing Run 10. The app was included in two studies in healthy participants and patients, focusing on binge eating disorder (n = 316) and major depressive depression (n = 65). Participants received a fixed compensation (€20) if they completed the online assessment. In the study on binge eating disorder, the online assessment including the app was part of a module to acquire data from participants with pathological eating behavior and binge eating episodes (subjective or objective). To ensure a wide range of symptoms of psychopathology, we did not exclude participants based on the presence of a mental disorder. Additional participants were recruited via the same channels such as social media, university mailing lists, and flyers without emphasizing the emphasis on symptoms of eating disorders or mood and anxiety disorders. Likewise, we invited participants contacting our lab to complete this online assessment including Influenca.

## Influenca and reinforcement learning game

To repeatedly assess reinforcement learning, we developed the cross-platform app Influenca. The app includes 31 runs of a reinforcement learning game based on a classic paradigm [49] with changing reward probabilities. Prior to each run, participants completed EMA items capturing momentary metabolic (hunger, satiety, thirst, time since last meal, consumption of coffee or snacks in the last two hours) and mental states (alertness, happiness, sadness, stress, distraction by environment, distraction by thoughts). Responses were given using either visual analog scales (VAS: hunger, fullness, thirst, alertness, happiness, sadness, stress, distraction by environment, distraction by thoughts), Likert scales (last meal), or binary scales (snack, coffee, binges).

In the Influenca app, a gamified version of a classic reinforcement learning task, participants had to fight a virus pandemic by finding the most effective medication and successfully treating as many people as possible (e.g., win as many points as possible). In each of the 31 runs, participants were presented with a new virus. Each run consisted of 150 trials and in each trial, they had to choose between two medications depicted as syringes of different colors with (initially) unknown win probabilities (Fig 1). At the beginning of each run, the colors of the options are randomly assigned to each side and the colors associated with the left and right syringe do not change within a run. Therefore, side and color contain the same information

**a**      **Screenshot of the app Influenca (exemplary level)**

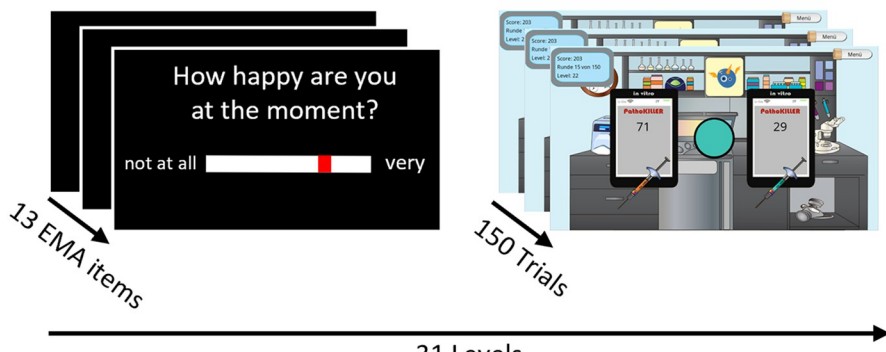

**b**      **Overview of task design**

**Fig 1. Illustration of the reinforcement learning task design. A**. Representative in-game screen of Influenca. To earn points, participants must identify which medication is most effective in fighting pathogens. In each trial, only one drug is effective to cure people. In this trial, the orange drug could treat 71 people and the turquoise drug would only treat 29 people. The circle depicts the color of the drug that was effective in the previous trial. If participants pick the correct medication, their score increases by the number of cured people (win). If they pick the incorrect medication, the number of falsely treated people will be subtracted from the score (loss). **B**. Procedure of the Influenca runs. Each run starts with ecological momentary assessment questions about participants' current mood and other states prior to the actual game, followed by 150 trials of the reinforcement learning paradigm. To ensure sampling across different states, there was a minimum of 2 hours waiting time enforced between the runs.

and learning can be associated to both domains. Notepads corresponding to each syringe showed the number of people cured in the trial, if the chosen medication was correct ("win"). A circle between the options depicts the color of the drug that was effective in the previous trial. After choosing one option, feedback about the choice was provided by either a green checkmark ("win") or a red cross ("loss") at the corresponding notepad. The corresponding points were added to the total score. If the chosen option resulted in a loss, the points were subtracted from the total score instead. A counter in the upper left corner showed the number of completed trials and the current total score.

In each trial, the number of points (reward magnitude) of each option was assigned randomly and added up to 100 across both options. Win probabilities were independent of reward

magnitudes and added up to 1 across both options. Since participants repeated the task up to 31 times, win probabilities of the options were determined by a Gaussian random walk algorithm and thus fluctuated over time. The use of random walks was intended to reduce meta-learning about when reversals or changes in contingencies would occur [50,51]. Each run was randomly initialized with a "good" ($p_{win}$ = 0.8) and a "bad" ($p_{win}$ = 0.2) option. To encourage replay, participants had to fight a new virus in each run. After completing a run, the defeated virus was added to a scoreboard and each completed run highlighted the scientists' increased "prestige" by showing an improved quality of the lab equipment, as depicted in the game's graphics.

### Experimental procedure

Participants installed the app on their preferred device by obtaining the installer file from our homepage (https://neuromadlab.com/en/influenca-2/). The app is available for Android, Windows, Linux, and MacOS. Participants provided a mail address at registration for app-specific communication (e.g., sending automated reminder mails, to send the individualized link for the questionnaires and the activation code to unlock runs 11 to 31 after completing the online questionnaires). To ensure confidentiality, the mail address was stored apart from the experimental data.

Before starting the first run, participants were asked to read through a detailed instruction explaining the controls as well as the game's cover story and rationale. They could re-read this instruction at any time by opening it via the game's menu. A version of this instruction was also posted in the download section on our lab homepage. Participants were instructed to play at different times throughout the day and in different (metabolic) states to sample data in diverse situations to improve generalizability. To ensure sufficient distinctiveness across runs, we required a delay between runs of at least 2 h, but there was no time restriction to complete the 31 runs. The data was stored locally on the participant's device and, once connected to the internet, synchronized with a database located at the Department of Psychiatry and Psychotherapy, University of Tübingen.

### Data analysis

#### Reinforcement learning model

To model different facets of reward learning, we used choice data from individual runs and fit a reinforcement learning model with two learning rates (α), reward sensitivity (β), and a parameter determining the additive weighting of reward magnitude and win probability during the choice. To ensure that our model was suitable for the data [52,53], the model was chosen after model comparison (for details, see S1 Appendix) between previously described models for this task [46,49,54]. Crucially, estimated parameters of the winning model showed better or similar reliability compared to the original model proposed by Behrens et al. [49], suggesting also an improvement in psychometric criteria (Table A in S1 Appendix). In the implemented reinforcement learning model, participants are assumed to decide between the options in each trial based on the inferred win probability of each option. These probability estimates, $p_{win}$, are learned, by updating the estimated win probability after each trial following a simple delta rule:

$$p_{win,t+1}(Option\ A) = p_{win,t}(Option\ A) + \alpha * RPE_t(Option\ A) \tag{1}$$

where $\alpha \in [0, 1]$ denotes the learning rate and the RPE the reward prediction error comparing

the expected win probability with choice outcome, that is scaled by the learning rate.

$$RPE_t(Option\ A) = r_t - p_{win,t}(Option\ A) \qquad (2)$$

$$r = \begin{cases} 1\ if\ Option\ A\ results\ in\ win \\ 0\ if\ Option\ A\ results\ in\ loss \end{cases} \qquad (3)$$

The learning rate therefore quantifies how quickly an individual updates choice preference with changing outcome contingencies. In other words, high learning rates lead to quick updates by putting more weight on recent choice outcomes. In contrast, low learning rates lead to slow updates by putting less weight on recent choice outcomes and more weight on former feedback, leading to smoother changes in estimated reward probabilities (Fig 2b).

In our winning model, we also included separate learning rates for wins and losses [54,55]. Next, choices in each trial are generated by combining the estimated win probability of the options ($p_{win}(Option\ A)$) with the associated reward values of each option ($f$) ($f_{OptA} \in [0, 100]$ and $f_B \in [0, 100]$ $with\ f_A = 100 - f_B$). In our winning model, both estimated win probability and reward magnitude are additively weighted [54]. This means that both the difference in win probability between the options and the difference in points between options are first scaled by a weighting parameter, $\lambda$, and then summed:

$$W_t(Opt.A) = \lambda(p_{win,t}(Opt.A) - (1 - p_{win,t}(Opt.A))) + (1 - \lambda)(f(Opt.A) - f(Opt.B)) \qquad (4)$$

Here, $\lambda$ is constrained between 0 and 1 and low values of the weighting parameter (Fig 2c upper panel) lead to decisions based on reward magnitude whereas high values (Fig 2c lower panel) lead to decisions based on win probability. Last, the probability to choosing option A is computed with a sigmoid probability function based on the weighted sum of win probability and reward points.

$$p_{choice,t}(Option\ A) = \frac{\exp(W_t(Option\ A)*\beta)}{(\exp(W_t(Option\ A)*\beta) + \exp(W_t(Option\ B)*\beta))} \qquad (5)$$

Here, the reward sensitivity, $\beta \in [0, Inf]$, scales the estimated weight of an option (i.e., weighted sum of win probability and points) with an individual factor indicating the subjective value. The reward sensitivity determines the predictability of value-based decisions in such a way that high reward sensitivity (Fig 2a, light blue colors have more extreme choice probabilities) leads to very predictable choices based on the weights, while low reward sensitivity (Fig 2a, dark blue colors have less clear choice probabilities) leads to noisier decisions [46]. This parameter is also referred to as inverse temperature and simultaneously captures choice stochasticity [9].

We fit the model using maximum likelihood estimation with the *fmincon* algorithm implemented in MATLAB 2020b. To ensure model parameters actually captured behavior [52], we assessed parameter recovery for the winning model by simulating data based on the estimated parameters for each run and applying the same model fit procedure to the simulated data. Parameters were successfully recovered with correlations between $r_{\alpha\_win} = .76$ and $r_{\alpha\_loss} = .94$ (Fig B in S1 Appendix).

To ensure that the choice behavior of participants was sufficiently well approximated by the model, we used the log-likelihood of each run as criterion. We excluded 106 runs (6.5% of all runs) with a poor model fit due to random decision-making (log-likelihood < -100.13). To determine this criterion, we fit simulated data with random choices for all trials and calculated the 95th percentile of the resulting log-likelihoods to include only runs that are improbable to

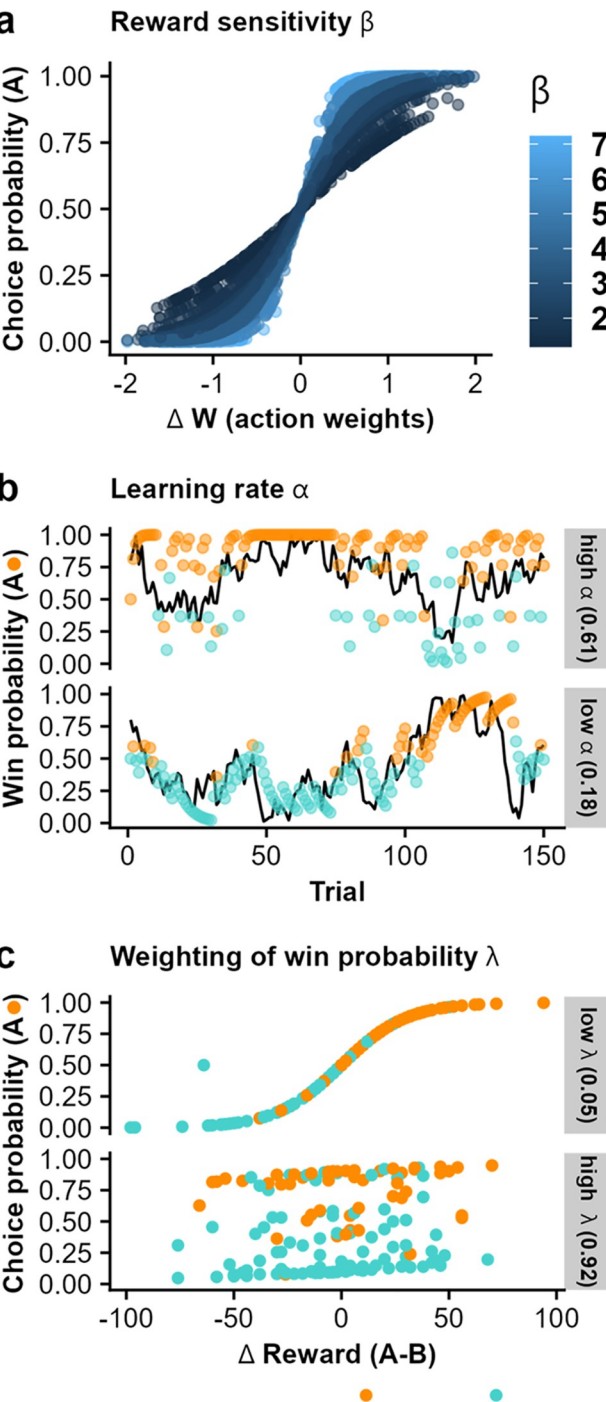

**Fig 2. Illustration of estimated parameters and their relation to differences in value-based decision-making and learning (representative participants). a.** The reward sensitivity beta scales how action weights (i.e., a combination of estimated probability and potential reward value) are translated into choices. Higher reward sensitivities translate to more deterministic choices (i.e., exploitation), whereas lower reward sensitivities lead to more random choices (i.e., exploration). **b.** The learning rate alpha captures how quickly estimated win probabilities are updated if new information is available. High learning rates (upper panel) lead to fast updates and quick forgetting of long-term outcomes. The black line depicts the latent win probability, while the points depict the estimated win probability based on the reinforcement learning model. **c.** Weighting of the estimated win probability of each option compared to the offered rewards is scaled by λ. Low values (< .5, upper panel) reduce the importance of the learned win probabilities

leading to choices based primarily on the potential reward at stake. In contrast, high values (>.5, lower panel) increase the importance of the learned win probabilities. Color in panel b and c indicates the observed choices in these exemplary runs.

arise solely from random choices. As model-independent performance measures, we used the average of earned points per trial for each run and response times for each decision. Trials with extreme response times (50 ms < response time < 10,000 ms) were excluded from the response time analysis (14.553 trials, 0.9% of all trials).

**Run effects.** To estimate effects of repeatedly playing the game on estimated parameters, we used linear mixed-effects models *lmerTest* [56]. We predicted the estimated behavioral parameters and model fit using the log-transformed run number as fixed effect. To account for inter-individual differences, run number and the intercept were modeled as random effects.

**Test-retest reliability.** To assess the reliability of behavioral parameters and state items, we estimated ICCs using linear mixed-effects models [57]. We report the ICC assessing absolute agreement as well as conditional ICCs considering systematic differences across runs (see S1 Appendix). To evaluate which runs within the game provide the most reliable parameter estimates, we calculated the test-retest rank correlation (Spearman) for each (held out) run with the average of the parameter across all other runs. We interpreted the correlations according to recommendations by Taylor [58] for correlation coefficients, where $r$s < .35 reflect low, $r$s between .36 and .67 reflect modest, $r$s > .67 reflect high correlations.

**Statistical threshold and software.** All statistical tests were performed using a significance level of $\alpha = 0.05$ (two-tailed). Data preprocessing was done with MATLAB 2020a. Linear mixed-effects models and ICCs were estimated in R Studio, R Version 3.5.3 [59]. Plots were created in R R Version 3.5.3 [59] using the package ggplot2 [60].

## Results

### Validation of parameter estimates and game mechanics

In each run, participants made 150 choices between two options of varying reward value and fluctuating win probabilities that had to be inferred over time. To estimate reinforcement learning parameters from value-based choices, we applied computational modeling. First, we analyzed which behaviors (i.e., parameter combinations) were associated with high average reward rates to determine optimal behavior. An increased average reward was reached with moderate learning rates for losses $\alpha_{loss}$ between 0.2–0.275 (all $p$s against other ranges, binned with .075 widths, < .05, Fig 3b). This indicates that not shifting from the preferred option in response to single negative outcomes is advantageous to track "true" fluctuations in the hidden random walks, as a high learning rate (Fig 2b, top panel) leads to fast switches in choices due to recent events. As found to be optimal in the task, a lower learning rate (Fig 2b, bottom panel) reflects more patience in light of surprising outcomes, reducing the number of switches between preferred options. In contrast, the learning rate for wins barely influenced the average reward (Fig 2c) as the task was designed to accommodate a large range of behavior without penalizing a given strategy. Moreover, our data shows a clear association of higher average rewards with higher reward sensitivities, although the improvement plateaus around the upper limit of our simulation ($\beta > 5$). Since reward sensitivity reflects the predictability of choices based on inferred differences in values, a high value indicates that successful participants make more deterministic choices even if there are only small relative differences in inferred value between the options. Last, making decision purely based on the difference in win probability while disregarding differences in potential reward as indicated by higher $\lambda$ values was

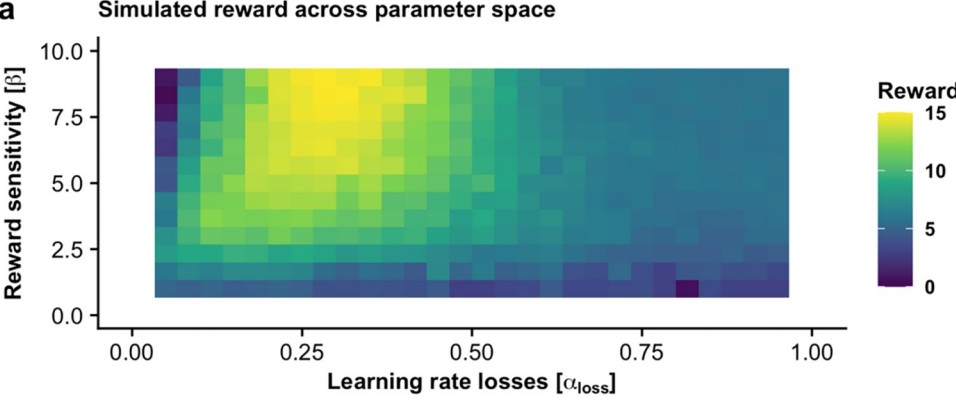

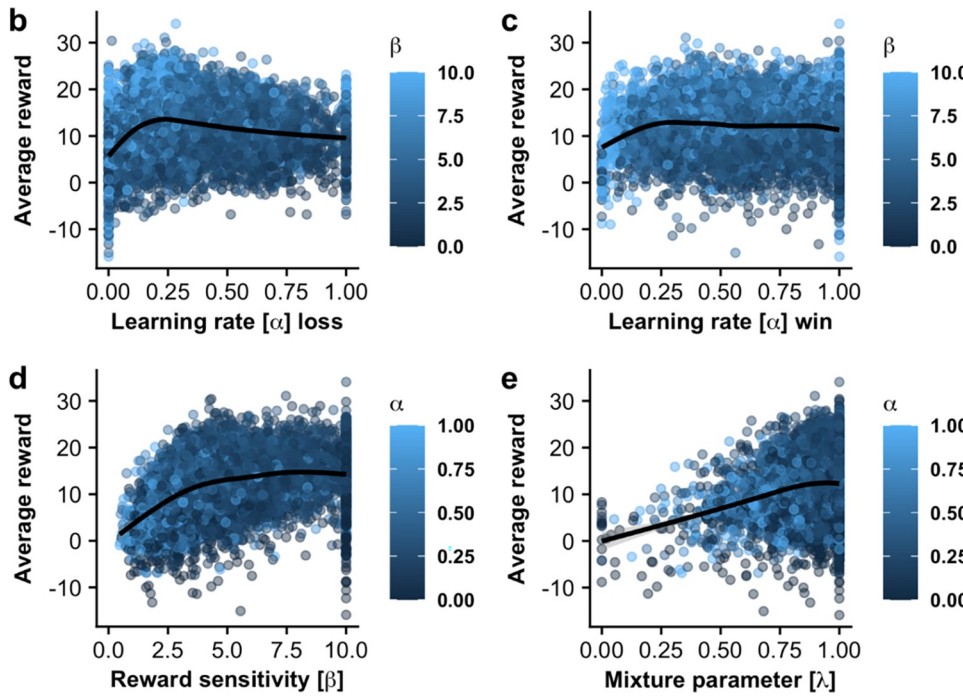

**Fig 3. Reward outcome per run for different combinations of learning parameters based on simulated and behavioral data. a**. Simulation of N = 50,000 players shows high rewards for different combinations of learning rate for losses and reward sensitivity. We show an exemplary grid for λ > .9, the optimal lambda (S3 Fig), and $α_{win}$ between .6 and .8, the average in our sample. Other combinations are shown in S3 Fig. **b**. Empirical data from the participants show high average rewards for moderate learning rates for losses between 0.2 and 0.275 (p < .05). **c**. Average reward is only weakly dependent on the learning rate for wins (r = -.04). **d**. Average reward increases with reward sensitivity (r = .32). **e**. Average reward is highest for lambda = 1 reflecting choices based on learned reward probabilities without considering reward values.

associated with the highest average rewards (Fig 3e) and almost all participants showed a very high λ.

## Reinforcement learning improves over runs

Second, we investigated changes in reinforcement learning over runs, to determine whether behavior converges to more optimal behavior. In general, participants successfully learned

the correct choice within the task as indicated by a positive average reward obtained (first run: $M = 8.98$, $SD = 5.85$; all runs: $M = 12.63$, $SD = 5.55$), and these rewards increased over runs ($t = 9.53$, $p < .001$). Repeatedly playing the game in addition led to decreased reaction times ($t = 20.03$, $p < .001$). The decrease in response times and increase in reward were negatively correlated ($r = -.25$; $p < .001$) suggesting that increased proficiency also speeds up decisions without a detrimental effect on accuracy. The improvement in model-independent performance indices was mirrored in the parameter estimates. Over runs, the learning rate for losses but not for wins decreased (Fig 4a–4b, loss: $t = -13.23$, $p < .001$, win: $t = -1.30$, $p = 0.19$) and the reward sensitivity ($t = 18.78$, $p < .001$), as well as weighting of estimated win probabilities ($\lambda$, $t = 9.74$, $p < .001$) increased (Fig 4c–4d). Moreover, the model fit (log-likelihood) also increased over runs (S4 Fig, $t = 22.19$, $p < .001$), suggesting that choices became more aligned with the estimated reinforcement learning model. Crucially, increased rewards were correlated (Fig 2 and Fig C in S1 Appendix) with an increased reward sensitivity ($r = .32$, $p < .0001$), weight on learned win probabilities ($\lambda$, $r = .23$, $p < .0001$), and model fit ($r = .52$, $p < .0001$) as well as decreased learning rates (alpha win: $r = -0.04$, $p < 0.001$, alpha loss: $r = -0.03$, $p < .0001$) indicating that participants converged to more successful strategies.

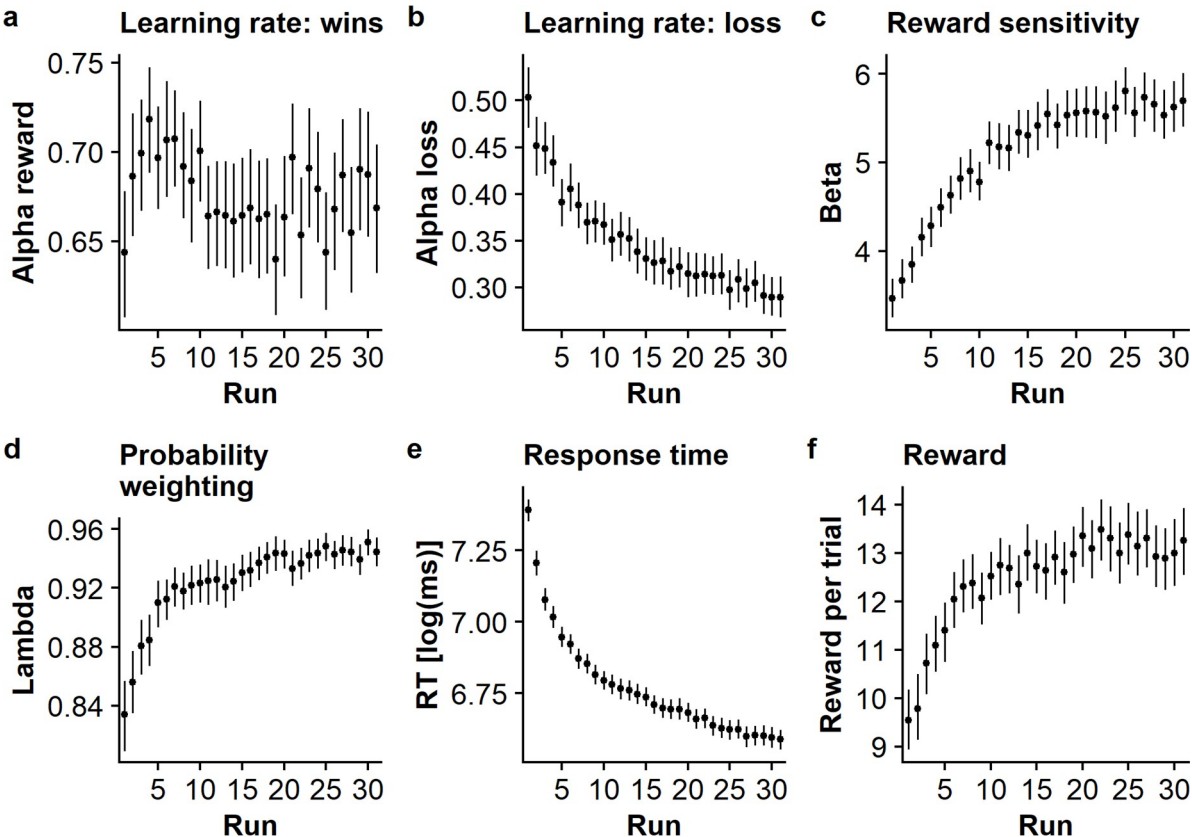

**Fig 4. Reinforcement learning parameters and model-independent performance indices improve over runs. a.** Learning rate for wins, $\alpha_{win}$, does not change over runs *(b = -0.006, p = .19)*. **b.** Learning rate for losses, $\alpha_{loss}$, decreases over runs *(b = -0.06, p < .001)*. **c.** Reward sensitivity $\beta$ increases over runs *(b = 0.72, p < .001)*. **d.** Weighting of win probabilities compared to reward magnitudes, $\lambda$, increases over runs *(b = 0.03, p < .001)*. **e.** Response times decrease over runs *(b = -0.23, p < .001)* **f.** The average reward increases over runs *(b = 1.09, p < .001)*. The dots show mean values with 95% bootstrapped confidence intervals.

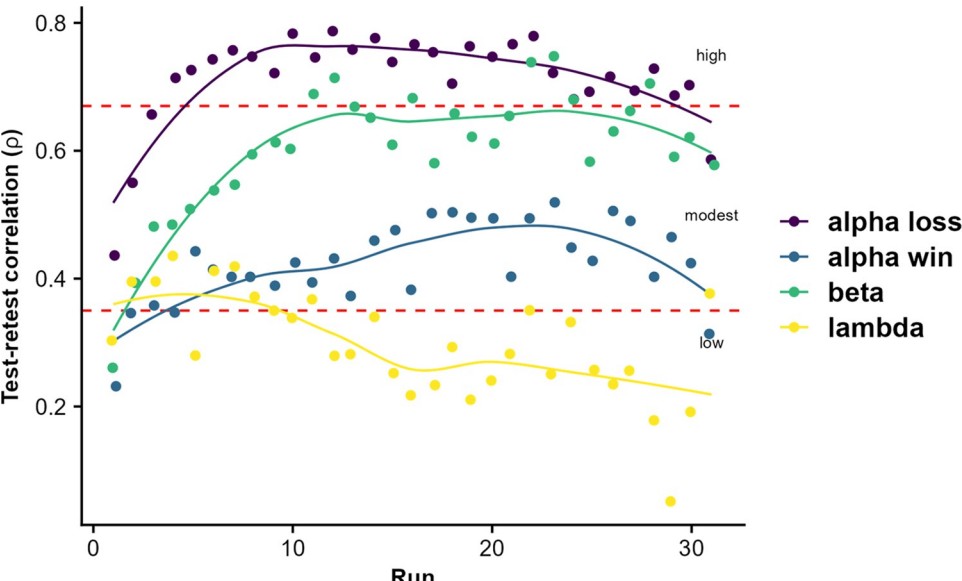

**Fig 5. Reliability of parameter estimates from the learning model improves after the first runs.** Dots depict the rank correlation of parameter estimates in one run with the mean across all other runs (leave-run out), separated per parameter. Red dashed lines show the classification of correlation magnitudes according to Taylor (1990). Results are independent of the exclusion criteria for behavior following other behavioral styles than reward learning as determined by extremely high log-likelihoods (S5 Fig).

## Reliability of reinforcement learning parameters is comparable to momentary states

Third, we assessed the reliability of reinforcement learning behavior, to determine their potential as individual biomarkers. ICCs of average rewards per run were poor (ICC = 0.10), indicating little grouping of data within individuals compared to overall variability across runs. Response times had higher ICCs compared to average reward rates (0.32). Model derived parameters yielded fair ICCs for learning rates for losses (ICC = 0.53) as well as weighting of win probabilities (λ: ICC = 0.40) but also poor ICCs for reward sensitivity (ICC = 0.34) and learning rate for wins (ICC = 0.23). Notably, ICCs were comparable (S2 Table) in both participants with vs. without depressive symptoms and participants with vs. without binge eating disorder and the ICC for weighting of win probabilities, λ, were even higher in groups with psychopathology (BDI> = 14: ICC = .43; BED: ICC = .55 vs. ICC .37 in participants without those symptoms). However, this increase in ICC for λ might also be explained by an increased between participant variability in those smaller subsamples, as in the complete sample many participants show similarly high values throughout all runs. To investigate whether early or late runs are more reliable, we calculated the Spearman rank correlation for each run with the average of all other runs. Notably, test-retest correlations of early runs were lower compared to late runs and improved until reaching a plateau after approximately 7 runs (Fig 5). This indicates that behavior within the task becomes more reliable with higher task proficiency, suggesting that late runs are better estimates of the "typical" performance on the task compared to early runs.

To relate the reliability of reinforcement learning parameters to the reliability of other measures (i.e., momentary states), we analyzed a subset of state items that are assessed prior to each run. We calculated ICCs of the items alertness, happiness, sadness, stress, distraction by environment, and distraction by thoughts (Table 1, Fig 6). The ICCs of EMA items ranged

**Table 1. Descriptive statistics and reliabilities of dependent variables.**

| Measures | Mean | SD | Median | 10th Percentile | 90th Percentile | ICC_unc | ICC_cond |
|---|---|---|---|---|---|---|---|
| **Behavioral indices** | | | | | | | |
| Response time [s] | 0.86 | 0.47 | 0.80 | 0.49 | 1.6 | .31 | .35 |
| Wins | 11.9 | 5.68 | 12.14 | 4.78 | 18.98 | .11 | .10 |
| **Model parameters** | | | | | | | |
| Log-likelihood | -41.6 | 20.9 | -40.5 | -70.2 | -15 | .42 | .40 |
| Learning rate win | 0.68 | 0.28 | 0.71 | .288 | 1 | .23 | .23 |
| Learning rate loss | 0.34 | 0.23 | 0.28 | 0.09 | 0.68 | .53 | .55 |
| Reward sensitivity | 4.98 | 2.37 | 4.44 | 2.41 | 9.4 | .33 | .35 |
| Lambda | 0.92 | 0.12 | 0.98 | 0.78 | 1 | .40 | .40 |
| **State items** | | | | | | | |
| Alertness | 56.8 | 22.8 | 58 | 26 | 87 | .30 | |
| Happiness | 59.1 | 23.3 | 60 | 26 | 90 | .40 | |
| Sadness | 29.3 | 24.9 | 23 | 1 | 67 | .41 | |
| Stress | 31.7 | 24.7 | 27 | 2 | 67 | .46 | |
| Distraction by environment | 25.7 | 23.4 | 19 | 1 | 62 | .35 | |
| Distraction by thoughts | 30.5 | 25.4 | 24 | 1 | 68 | .52 | |

*Note*: SD = standard deviation, ICC = intraclass correlation coefficient

from poor (alertness: 0.30, distraction by environment: 0.35) to fair (sadness: 0.41, happiness: 0.40, stress: 0.46, distraction by thoughts: 0.52). Of note, ICCs were in a range comparable to the reinforcement learning parameters. Illustratively, items reflecting emotional states (happiness, sadness, stress) had similar ICCs as the loss learning rate, the most reliable reinforcement

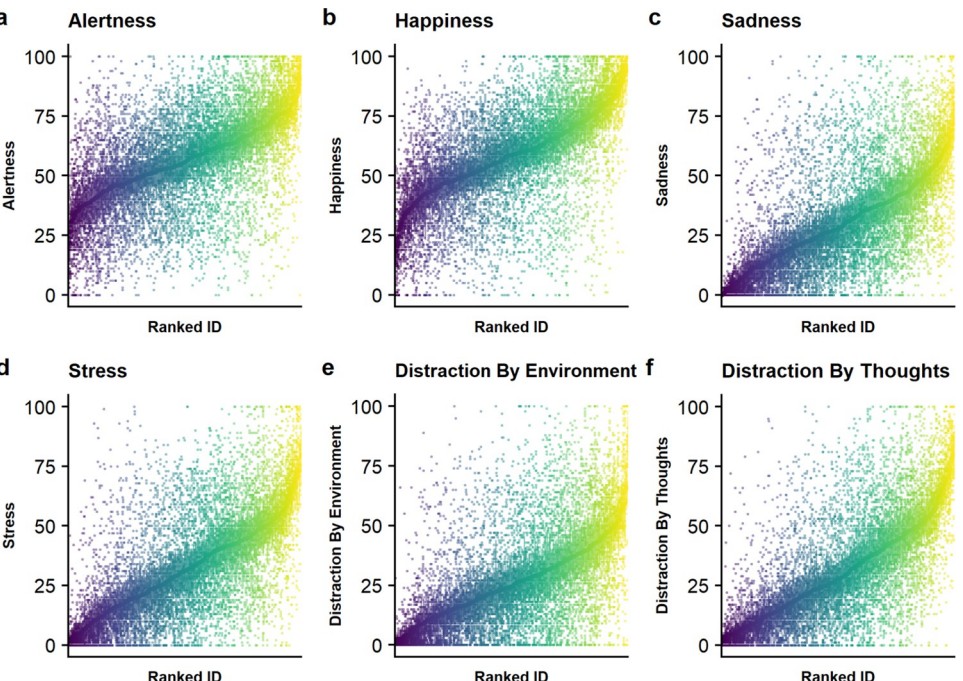

**Fig 6. State items show poor to fair intra-class correlation coefficients *(0.29–0.53)*. A-F.** Mean and variance of selected EMA items per individual, ranked by mean value of each participant across runs. Dots represent the values per run.

learning parameter. These results point to a comparable ratio of within- and between-subject variance for behavioral parameters as for alleged state items.

## Discussion

Value-based decision-making and learning are integral parts of adaptive behavior. Since alterations have been linked to multiple mental disorders, it is pivotal to develop accessible tools that capture reliable inter-individual differences in value-based decisions to advance the use of deeply phenotyped behavioral data for clinical classification and prediction. To this end, we introduce our open-source cross-platform application Influenca, which comprises multiple runs of a reward learning task and (customizable) EMA state items. Using preliminary data of 9729 runs from users with a minimum of ten runs, we show that our app provides detailed insight into the reliability of common indices of value-based decision-making and learning over extended periods of time. In line with the few available reports of test-retest reliability in comparable lab-based assessments, the reliabilities of the estimated model parameters were poor to fair, suggesting that learning parameters fluctuate substantially over runs. Such fluctuations may limit the prospect of using single runs for individual diagnostics. Likewise, the reliability of single-run estimates increased after several runs, highlighting the benefit of multi-run assessments to improve psychometric properties. Taken together, future use of innovative tools such as Influenca in large-scale, naturalistic assessments may provide a much more nuanced perspective on individual trajectories in decision-making and learning and their contribution to mental health.

Influenca features several key innovations to study value-based decision-making and learning at scale as part of longitudinal studies in a naturalistic setting. Since the app was made available on the participants' preferred devices and was completed outside of a controlled laboratory setting, a careful evaluation of the data quality is essential [14,61,62]. To validate the online assessment, we show that participants quickly learn to do the task well and perform it in a moderately reliable manner after a few runs (e.g., ~ 10 runs are necessary before the reward sensitivity estimates become moderately reliable). Consequently, players win more points than expected by chance in most of the runs 98% (score > .33; highest average win per trial when choices are simulated randomly) and wins increase over runs, indicating that participants played the task with increasing proficiency. In parallel with improvements in learning across runs, response times decreased, indicating that participants speed up their deliberation process with increasing proficiency as well. Beyond basic performance indices, we observed that participants showed changes in reinforcement learning parameters over runs. For example, the average learning rate for losses decreased over runs, indicating an integration of feedback about win probabilities across more consecutive decisions. In contrast, reward sensitivity increased over runs, reflecting greater exploitation of learned contingencies with increased task proficiency [63–65]. By design, Influenca does not promote a narrow range of learning rates but captures individual differences in value-based decision-making over the course of the game. Such inter-individual variance is crucial for the effective use of tasks in precision psychiatry [13]. To conclude, our newly developed app captures differences in inter-individual and intra-individual decision-making in a naturalistic setting and tracks increased proficiency and test-retest reliability as reflected in behavioral estimates over time.

Based on our extensive data of repeated runs of the task, we provide a more refined insight into the test-retest reliability of value-based decision-making and learning. Across all runs, the reliability of behavioral indices was poor to fair, indicating a limited trait-like characteristic of value-based decision-making [46]. However, the observed ICCs of the learning rate are in accordance with previous studies using lab-based assessments [34,36,37] suggesting it is

unlikely to be caused by the naturalistic setting. Notably, performance as well as model-fit improved over runs, indicating that participants became more proficient and behaved more in line with the assumed RL model. To see whether increased task proficiency would lead to improved reliability, we correlated estimates of single runs with the average of the held-out runs. Crucially, reliability of the learning rate and reward sensitivity increased up to run 7, suggesting that initial variance in the parameter estimates is not necessarily as predictive of trait-like differences in learning as late variance in the presence of substantial task expertise. Therefore, a certain amount of training might be necessary before reliable learning estimates can be derived from RL tasks. Likewise, multi-run assessments conducted across various mental or metabolic states (S6 Fig for distributions) may provide a better approximation of generalizable inter-individual differences in value-based decision-making compared to the typical single-run assessments in the lab after limited practice on the task.

Despite its notable strengths, our study has several limitations that should be addressed in future research. First, in comparison to lab-based experimental setups, we cannot control the testing environment our participants are confronted with when they interact with the app. Still, we argue that the naturalistic setting of EMA has important advantages that can outweigh the limited control over standardized data collection, such as collecting more data points that are representative of participants' daily lives. Second, to prevent learning of underlying task structures and thus enabling repeated measurements for up to 31 runs, we implemented randomly fluctuating reward probabilities for both options. Therefore, behavior will vary across runs even in response to idiosyncrasies of single runs. Arguably, this is a trade-off in repeated measurements of reinforcement learning that influences test-retest reliability. Still, systematic changes can be addressed in hierarchical models to provide a good reference for interventions. Third, to illustrate the rationale of our app, we chose a basic set of reinforcement learning models suggested in the seminal work by Behrens and colleagues [49] and later work by Gagne et al [54]. More advanced models, for example, incorporating reaction times [44], could provide deeper insights into the decision-making processes and their progression over repeated runs. Potential advances include a hierarchical Bayesian framework [66], allowing to track the agent's estimate of the current volatility in the environment, and asymmetric learning rates for wins and losses [67]. Moreover, in our model fitting, we did not incorporate the nested data structure that is inherent in repeated measures and not accounted for when all runs are considered as independent. We derived single-session estimates that, in line with classical ICCs, provide psychometric properties of task measures that are frequently used without generative modeling or when only data from one session is available [68] as this corresponds to the intention of using an instrument in a given diagnostic application. Arguably, this leads to lower reliabilities compared to a full Bayesian model that capitalizes on the information conferred by other sessions [42,43], but this also alters the interpretation of derived reliability indices. Incorporation of priors in Bayesian models leads to shrinkage of parameters which increases reliability of parameters but at the same time may also reduce meaningful variability within participants that is not noise but related to fluctuations in other states [45]. Still, this could be exploited with Bayesian fitting methods that have been shown to improve test-retest reliability of model parameters [41,43,69] as well as the estimation of trait-like characteristics of behavior [70]. Fourth, apart from the rich possibilities provided by extensions of the computational models, future work should focus on the reciprocal influence of momentary states (including metabolic states such as hunger) and parameters of value-based decision-making [17,71] to differentiate meaningful fluctuations in reward learning behavior from measurement noise that leads to reduced reliability. To this end, hierarchical Bayesian models can be extended to directly incorporate the metabolic or mood state of each run to then modulate RL parameters [72]. Last, to provide a gamified version of the task, we chose a framework conveying the basic

mechanics of the task such as changing reward probabilities corresponding to changes in effectiveness of medication or additional levels associated with new pathogens that have to be treated. The design of the task was chosen considerably before onset of COVID-19 and even data collection started before the COVID-19 pandemic. Consequently, behavior might have changed considering the gamified content of fighting different viruses. Since the majority of participants started playing after the onset of the pandemic and reliability primarily evaluates potential changes in the rank order of participants, systematic effects of the pandemic on psychometric properties should be small in comparison other sources of variance.

To summarize, online and smartphone-based assessment has gained traction as a scalable method for longitudinal studies in larger and more representative samples embedded in a naturalistic setting that may improve generalizability. Here, we provide a psychometric evaluation of our open source, cross-platform reinforcement learning task for future use in large-scale assessments of individual differences in value-based decision-making. We show that our gamified task captures inter-individual and intra-individual differences in decision-making and learning, which can be associated with naturally occurring fluctuations in state measures or capitalized to evaluate behavioral effects of interventions. Consequently, this may provide more nuanced insight into behavioral changes. Based on our extensive longitudinal assessment of reinforcement learning, we provide detailed information on the test-retest reliability of behavioral performance indices and model parameters, suggesting that multiple runs per participant are necessary to provide sufficient diagnostic information at the individual level. Furthermore, later runs of the game show better model fit and higher reliability, indicating that greater task proficiency improves the estimation of parameters that better reflect stable behavioral traits. To conclude, our reinforcement learning task can be used to precisely track the dynamics of value-based decision-making and learning providing a new avenue for future research to improve the individualized prediction of behavior as well as the classification of individuals for diagnostic or clinical purposes.

## Supporting information

**S1 Fig. Maximum number of completed runs per participant in our updated sample.**
(TIF)

**S2 Fig. Distribution of the Log-likelihoods derived from fitting all runs included in the sample with simulated random choices.** We only included runs with a low chance of coming from random choices (i.e., Log-likelihoods higher than the 95 percentile from this random distribution, -100.13).
(TIF)

**S3 Fig. Simulated average rewards across the parameter space show that moderate learning rates for losses together with high reward sensitivities and mixture parameters lead to highest rewards.** The learning rate for wins has less influence on the obtained rewards apart from very low learning rates.
(TIF)

**S4 Fig. The Log-likelihood of the models increased over runs indicating that later runs showed less noisy behavior that was more in line with the computational reinforcement learning model.**
(TIF)

**S5 Fig. Reliability is not influenced by including or excluding runs with extreme model fits. Since runs with very high Log-likelihoods (between 0 and -10) were characterized by**

**many boundary estimates we calculated test-retest correlations across different runs for different exclusion criteria.** The patterns across runs and between parameters remain comparable across all exclusion criteria.
(TIF)

**S6 Fig. Participants completed Influenca levels across a variety of metabolic states.** a)-b) distribution of individual ranges (i.e., maximal value–minimal value) of hunger (a) and satiety (b) ratings show that more than 90% participants had ranges exceeding 50 or 60 points for hunger and satiety, respectively. c)-d) Density of run-based hunger and satiety ratings for each participant show that large areas of possible metabolic states are covered in most participants.
(TIF)

**S1 Table. Demographic and psychometric information on the sample.**
(DOCX)

**S2 Table. Reliability measures for participants with BDI > = 14 vs. BDI < 14 and participants with vs. binge eating disorder.**
(DOCX)

**S1 Appendix. Detailed description of model validation including model comparison, parameter recovery, model formulas, parameter correlations and ICCs.**
(PDF)

## Acknowledgments

We thank Gizem Altan and Anastasia Illarionova for providing the graphics for Influenca, and Jennifer Them for support in programming. We thank Hannah Schütt for help in setting up the database. We thank Franziska Müller, Magdalena Ferstl, Salome Herwerth, Dana Wentz for help with data acquisition as well as Wiebke Ringels and Johanna Theuer for support in additional analyses.

## Author Contributions

**Conceptualization:** Jennifer Svaldi, Nils B. Kroemer.

**Data curation:** Monja P. Neuser.

**Formal analysis:** Monja P. Neuser, Anne Kühnel, Franziska Kräutlein, Nils B. Kroemer.

**Funding acquisition:** Jennifer Svaldi, Nils B. Kroemer.

**Methodology:** Anne Kühnel, Vanessa Teckentrup, Nils B. Kroemer.

**Project administration:** Monja P. Neuser.

**Software:** Vanessa Teckentrup.

**Supervision:** Jennifer Svaldi, Nils B. Kroemer.

**Visualization:** Anne Kühnel, Franziska Kräutlein.

**Writing – original draft:** Monja P. Neuser, Anne Kühnel, Franziska Kräutlein, Nils B. Kroemer.

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
