## [Decision Letter · Decision Letter 0]

28 Apr 2023

PDIG-D-22-00326

Influenca: a gamified assessment of value-based decision-making for longitudinal studies

PLOS Digital Health

Dear Dr. Kroemer,

Thank you for submitting your manuscript to PLOS Digital Health. After careful consideration, we feel that it has merit but does not fully meet PLOS Digital Health's publication criteria as it currently stands. Therefore, we invite you to submit a revised version of the manuscript that addresses the points raised during the review process. Please see additional editorial comments below the email signature.

Please submit your revised manuscript within 30 days May 28 2023 11:59PM. If you will need more time than this to complete your revisions, please reply to this message or contact the journal office at digitalhealth@plos.org. Please include the following items when submitting your revised manuscript:

We look forward to receiving your revised manuscript.

Kind regards,

Grigorij Schleifer, MD

Guest Editor

PLOS Digital Health

Nitika Pant Pai

Section Editor

PLOS Digital Health

Journal Requirements:

2. We ask that a manuscript source file is provided at Revision. Please upload your manuscript file as a .doc, .docx, .rtf or .tex.

3. Please provide separate figure files in .tif or .eps format only and remove any figures embedded in your manuscript file. Please also ensure that all files are under our size limit of 10MB.

4. We have noticed that you have uploaded Supporting Information files, but you have not included a list of legends. Please add a full list of legends for your Supporting Information files after the references list.

Additional Editor Comments (if provided):

Dear Dr. Kroemer,

This is an interesting and important work.

I have only a few comments, and most of the points have been addressed by the reviewers. As both reviewers have pointed out, the mathematical aspects of the publication should be made more accessible to a broader audience. It took me several attempts to read the publication and understand the mathematical background of different reinforcement learning models and their importance for this study.

I am curious to know which patient population would benefit from the game?

Additionally, the legend of Figure 3 is misleading, and it is not clear which colors represent A or B.

Could you please provide the R or MATLAB code for the analysis?

Please implement the comments and resubmit.

Thank you.

Reviewers' comments:

Reviewer's Responses to Questions

**Comments to the Author**

1. Does this manuscript meet PLOS Digital Health’s publication criteria? Is the manuscript technically sound, and do the data support the conclusions? The manuscript must describe methodologically and ethically rigorous research with conclusions that are appropriately drawn based on the data presented.

Reviewer #1: Yes

Reviewer #2: Yes

2. Has the statistical analysis been performed appropriately and rigorously?

Reviewer #1: Yes

Reviewer #2: Yes

3. Have the authors made all data underlying the findings in their manuscript fully available (please refer to the Data Availability Statement at the start of the manuscript PDF file)?

Reviewer #1: Yes

Reviewer #2: Yes

4. Is the manuscript presented in an intelligible fashion and written in standard English?

Reviewer #1: Yes

Reviewer #2: Yes

5. Review Comments to the Author

Reviewer #1: Abstract:

1. (minor) "Here, we developed..." -> "In this paper we present..."

2. (minor) "Notably, state items showed comparable ICCs as reinforcement learning parameters." -> It is not clear to me what the term "state items" refers to.

Method:

3. "Additional participants were recruited through social media, university mailing lists, and flyers. Moreover, participants in other studies of the lab are invited to play Influenca." -> How many participants were recruited from each population? Is it possible to evaluate what percentage of each population (i.e., mental disorders, healthy, etc.) played more than 10 runs and were therefore included in the analysis?

4. The "virus"/"pandemic" theme might have lead to differences in participants' behavior before and after the outbreak of the Covid-19 pandemic. Consider analyzing potential differences in participants behavior before 2020 and after 2020. If this is impossible, critically evaluate the potential influence in the limitations section.

5. Reinforcement learning model: (p. 14) I understand why you would perform the following analyses based on the “best” model (i.e., the one which you identified beforehand). However, it might be worth considering to use one or a few alternative models in addition to the seemingly best one. I think that this would help to proof that the identified model really is the best one. One would not even have to report the analytical procedure of the alternative models, but rather state that all analyses have been run with different models (and model parameters), and the model with the two learning rates and the additive weighting performed best in all analyses.

6. Test-retest-reliability: This section seems rather long and elaborate, as test-retest-reliability is a common and well-known statistical test. I would recommend shortening the paragraph.

Results:

[none]

Discussion:

7. “Since alterations [of value-based decision-making] have been linked to multiple mental disorders [...]” → The participants suffer, at least in part, from different mental disorders (as described in the method section). However, the inter-group differences between healthy participants and participants with binge eating disorders or depression have not been investigated. Would it not be interesting to see if the three “groups” differed regarding the quality of their value-based decision-making?

8. (minor) “to boost psychometric properties” → While I like the enthusiastic connotation, I would recommend choosing another term.

9. (minor, general) "Run-based data that support the findings of this study will be publicly shared on osf

upon publication." While this might be a standard phrase to inform reviewers/readers about data availability, it sounds as if the authors only share data that supports the findings of the study while keeping conflicting findings to themselves. I am sure that this is not the case, but it might be a good idea to reword this sentence.

Reviewer #2: The manuscript by Neuser et al. provides a valuable insight into valuable-based decision-making and its measurement methods. Nevertheless, I would like to suggest some ideas and questions for further improvement. All suggestions are grouped according to the respective section. 

Introduction: 

- The plethora of statistical parameters of the referenced studies could be omitted for the sake of conciseness

- The content in this section could be more balanced in terms of reliability and psychological elements (e.g. psychobiological process, biomarkers). I would suggest to reduce the parts for reliability to eventually strengthen the conciseness 

Methods: 

- It is not clear why participants with binge eating disorder and depression, as well as healthy participants could join the study. This should be explained, maybe with the help of mentioning further exclusion criteria. Furthermore, it should be stated why different groups of participants could join the study and why these subgroups were not evaluated respectively. 

- Why had participants to play at least 10 runs to be included? 

- Why was the storyline / narrative of Influenca, a clinical topic, chosen? Especially since participants with psychiatric disorders were included in the study. Is there a coherence between these domains? 

- Are there two versions of the app? The description of the app gives the impression that there is one gamified version and one version without game design elements.

- I would suggest introducing more precisely, why it was important to create different reinforcement learning models. Is it relevant for the statement of the paper to mention the models in such detail? Does it fit with the remaining sections? 

- Do the learning models include the states captured via EMA? 

Results: 

- Both sections, results and methods, include sophisticated content and might not be easy to understand for the broader audience. As the journal addresses the broader audience, I would suggest simplifying these sections. 

- Thought could be given to waiving the detailed description of creating the models in favor of embedding the results into the broader context of psychiatric disorders with a reference to the study population. Especially as the discussion starts with a reference to psychiatric disorders. 

Discussion:

- It should be discussed why the storyline of Influenca is used for participants with psychiatric disorders (as already mentioned in the methods section)

- A latency of at least 2 hours was stated to secure for different metabolic states, but can this be guaranteed? Was it somehow controlled? Otherwise, this should be discussed.

6. PLOS authors have the option to publish the peer review history of their article (what does this mean?). If published, this will include your full peer review and any attached files.

**Do you want your identity to be public for this peer review?** For information about this choice, including consent withdrawal, please see our Privacy Policy.

Reviewer #1: Yes: Matthias Carl Laupichler

Reviewer #2: No

---

## [Editor Report · Decision Letter 1]

17 Jul 2023

Reliability of gamified reinforcement learning in densely sampled longitudinal assessments

PDIG-D-22-00326R1

Dear Dr. Kroemer,

we are pleased to inform you that your manuscript 'Reliability of gamified reinforcement learning in densely sampled longitudinal assessments' has been provisionally accepted for publication in PLOS Digital Health.

Best regards,

Grigorij Schleifer, MD

Guest Editor

PLOS Digital Health